# Structural and Geomechanical Analysis of Natural Caves and Rock Shelters: Comparison between Manual and Remote Sensing Discontinuity Data Gathering

**Abdelmadjid Benrabah** [1], **Salvador Senent Domínguez** [1], **Fernando Carrera-Ramírez** [2], **David Álvarez-Alonso** [3], **María de Andrés-Herrero** [3] **and Luis Jorda Bordehore** [1,*]

[1] ETSI Caminos, Canales y Puertos, Universidad Politécnica de Madrid, C/Prof. Aranguren, s/n, 28040 Madrid, Spain; abdelmadjid.benrabah@alumnos.upm.es (A.B.); s.senent@upm.es (S.S.D.)

[2] RAC, Rock Art Conservation and Management, 36202 Vigo, Spain; rac.fcarrera@gmail.com

[3] Department of Prehistory, Ancient History and Archaeology, Complutense University of Madrid, C/Profesor Aranguren, 28040 Madrid, Spain; david.alvarez@ucm.es (D.Á.-A.); maria.deandres@ucm.es (M.d.A.-H.)

\* Correspondence: l.jorda@upm.es

**Abstract:** The stability of many shallow caves and rock shelters relies heavily on understanding rock discontinuities, such as stratification, faults, and joints. Analyzing these discontinuities and determining their orientations and dispersion are crucial for assessing the overall stability of the cave or shelter. Traditionally, this analysis has been conducted manually using a compass with a clinometer, but it has certain limitations, as only fractures located in accessible areas like the lower part of cave walls and entrances are visible and can be assessed. Over the past decade, remote sensing techniques like LiDAR and photogrammetry have gained popularity in characterizing rocky massifs. These techniques provide 3D point clouds and high-resolution images of the cave or shelter walls and ceilings. With these data, it becomes possible to perform a three-dimensional reconstruction of the cavity and obtain important parameters of the discontinuities, such as orientation, spacing, persistence, or roughness. This paper presents a comparison between the geomechanical data obtained using the traditional manual procedures (compass readings in accessible zones) and a photogrammetric technique called Structure from Motion (SfM). The study was conducted in two caves, namely, the Reguerillo Cave (Madrid) and the Cova dos Mouros (Lugo), along with two rock shelters named Abrigo de San Lázaro and Abrigo del Molino (Segovia). The results of the study demonstrate an excellent correlation between the geomechanical parameters obtained from both methods. Indeed, the combination of traditional manual techniques and photogrammetry (SfM) offers significant advantages in developing a more comprehensive and realistic discontinuity census.

**Keywords:** remote sensing techniques; structure from motion (SfM); geomechanical analysis; caves and shelter; photogrammetry

## 1. Introduction

The stability of slopes and underground excavations is controlled by various structural features, such as stratification, faults, and joints [1–4], as well as rock properties and weathering. Furthermore, a proper classification of discontinuities is key to assessing the stability of caves and shelters [5]. In general, geotechnical experts ascertain the mechanisms of deformations and quantities of folds in rock using landscape investigation to collect and document the orientation data of the rock mass using a compass equipped with a clinometer, scanline survey, and measuring tape [6–8]. In recent decades, the use of remote sensing analysis and digital geological surveys has provided more information regarding various rock mass discontinuities [9,10].

The manual technique using a compass is still widely employed, but it possesses constraints as the data collection process relies on visual inspection and manual quantification,

which are contingent on the expertise and assessment of the engineers and geologists [11,12] and the accessibility of the sites. For example, large caves present important problems related to access to roofs and walls for collecting structural data, and only in lower areas can the geomechanical characterization be carried out. Frequently, this tends to result in biased outcomes and, consequently, inadequate evaluations of hazard susceptibility ratings and the potential risks associated with unstable rocks [13,14].

The advancement of remote sensing imaging technologies (terrestrial laser scanning (TLS) and photogrammetry) has facilitated the acquisition of 3D data of the terrain surface with exceptional precision in a reduced timescale and has provided expanded geographical coverage [15–17].

The two main remote data collection techniques are Interferometric Synthetic Aperture Radar (InSAR) and Light Detection and Ranging (LiDAR) [16,18]. InSAR enables high-precision measurements of terrain surface movements to be obtained [15], which are primarily utilized for landslide monitoring and detection [19]. On the other hand, LiDAR captures a 3D point cloud representation of the terrain surface, which offers the possibility of working with geometrical surface information. The comparison between point clouds acquired during different time frames also enables the monitoring and detection of landslides [20,21].

In addition to the previous remote data collection techniques, digital photography has experienced substantial progress through the evolution of photogrammetry methods. One such technique is Structure from Motion (SfM), which enables the reconstruction of surfaces using 3D point clouds derived from digital photographs. SfM is recognized as an automated, high-resolution, and cost-effective photogrammetry approach rooted in the principles of stereoscopic photogrammetry (reconstructing 3D structures from image superposition). Originally stemming from artificial vision and the development of automated algorithms for digital image correlation (DIC), SfM deviates from conventional photogrammetry by automatically solving scene geometry, camera positions, and orientations without the need to establish a pre-defined control point network with known 3D coordinates. The process involves solving collinearity equations from many conjugated points (common image points) identified during the automatic correlation phase of a series of superimposed images acquired in an unstructured manner [22,23]. Compared to LiDAR instrumentation, the equipment used in SfM incurs lower economic costs while still yielding reasonably acceptable results [24,25]. The limitations of SfM depend on factors such as the lens quality, processing time, image-capturing procedures, and machine resource consumption.

In the context of underground spaces, employing digital photogrammetry presents challenges due to the dark environment, necessitating meticulous preparation for image capture and careful consideration of lighting conditions. Nevertheless, the quality and resolution of the images directly influence the resulting model's quality. A greater number of high-quality images contributes to superior model outputs. However, this also means increased resource consumption and a longer computational time [15,26].

In recent years, several caves and shelters have been documented using TLS (Terrestrial Laser Scanning) and LiDAR (Light Detection and Ranging) for various applications [27]. However, the use of Structure from Motion (SfM) in caves and rock shelters for stability assessment has been rarely reported [28]. As a result, this study aims to compare and combine both manual data collection and photogrammetry SfM methods to validate SfM's applicability in assessing the stability of caves and rock shelters.

## 2. Regional Framework

The studied sites are two caves and two shelters that have relevant historical and archaeological value and are located in different zones of Spain (Figure 1).

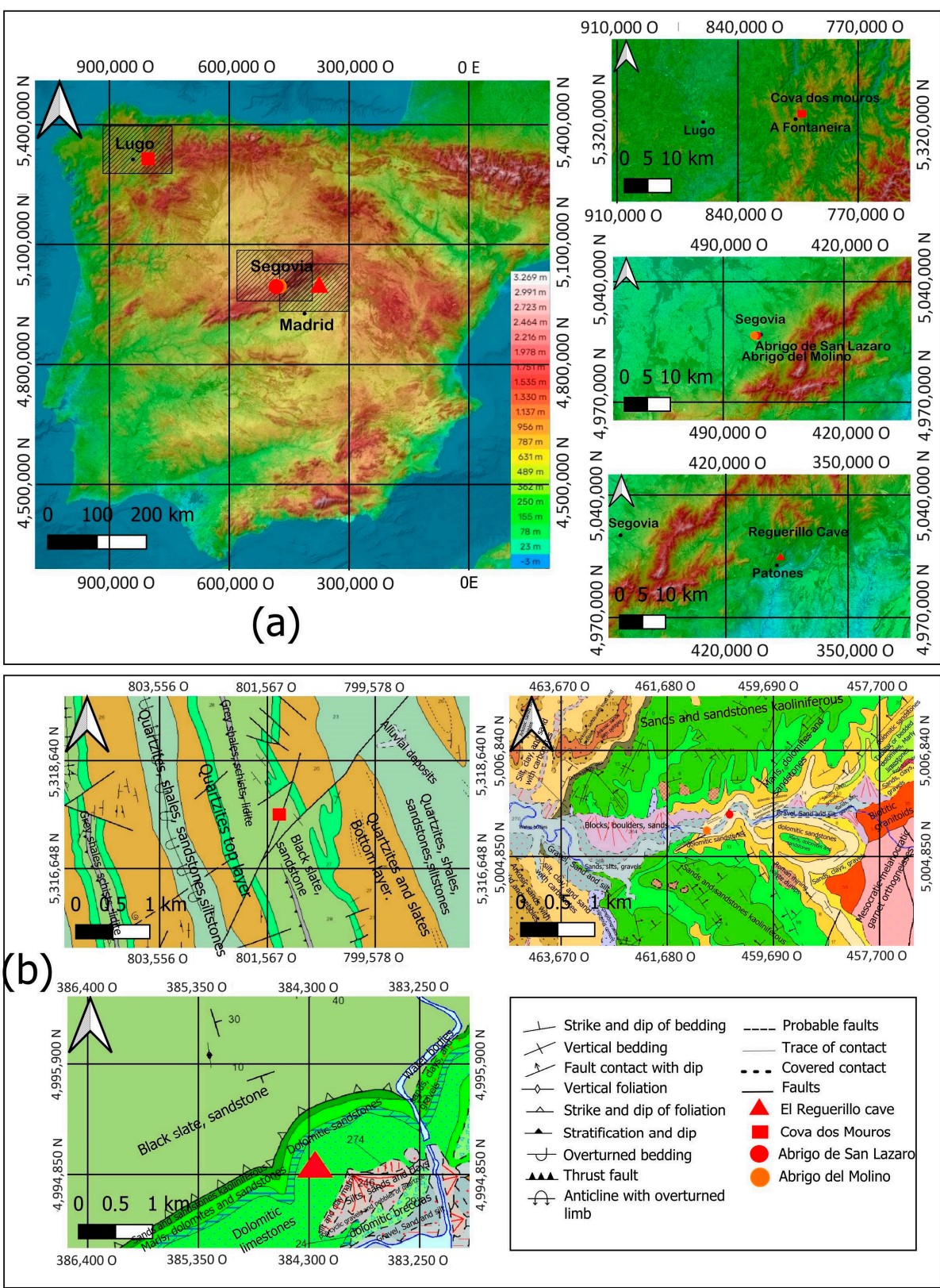

**Figure 1.** Location and geological context: (**a**) location of the study sites; (**b**) geological setting of the study area.

The Reguerillo Cave is located in Patones, a small village in the north of Madrid province. This cave (Figure 2a) is a reference point for speleology in Madrid and in the entire central area of Spain [29,30]. It is in the Cretaceous band that borders the

Guadarrama–Somosierra–Ayllón Mountains ranges formed by dolostones, dolomitic sandstones, limestone and marly limestone. The cave has a length of 8 km. Geomechanical characterization in the field campaign was only focused in the southern cave entrance (Figure 2a) because the entrance is closed and forbidden due to its archaeological relevance.

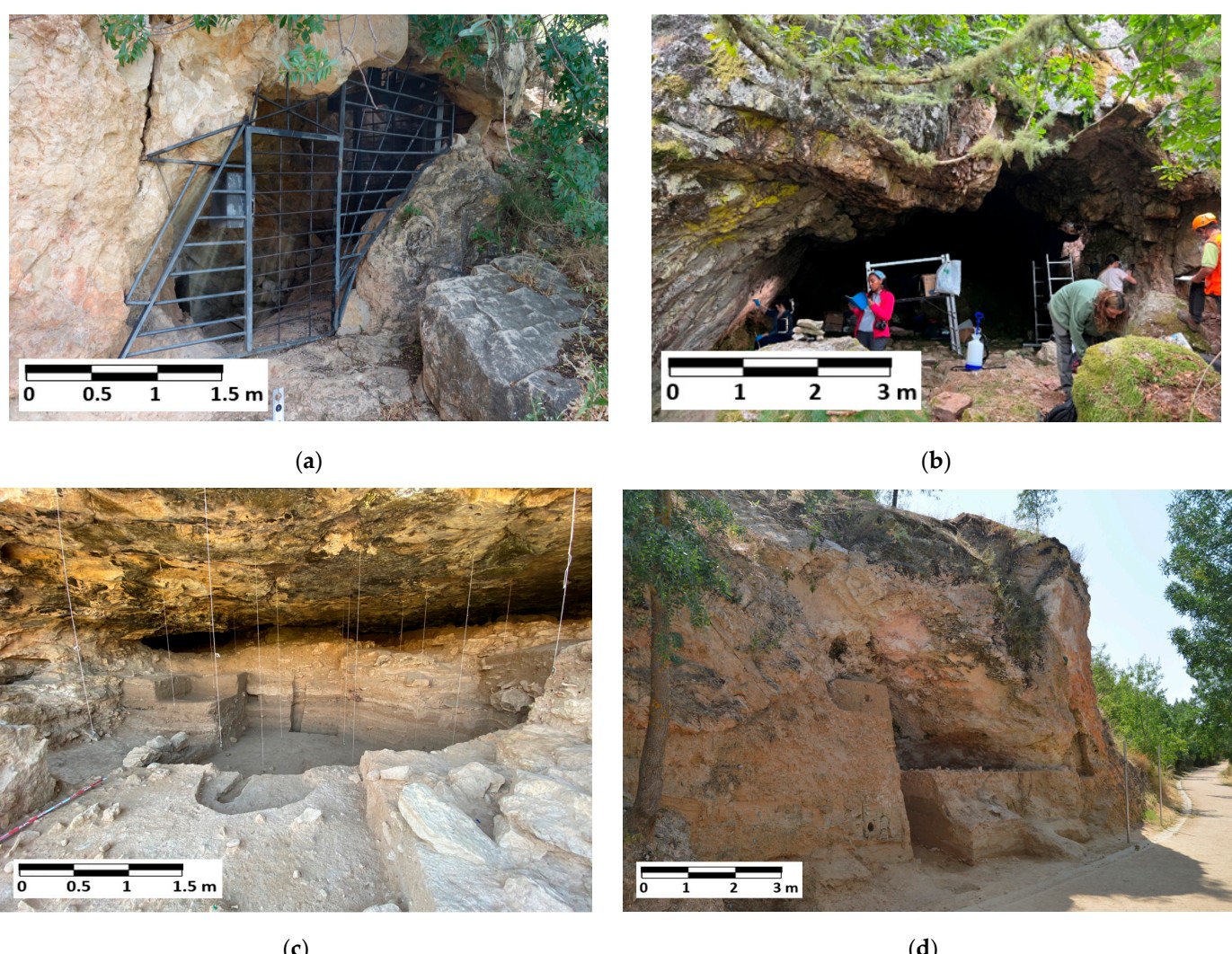

(a)

(b)

(c)

(d)

**Figure 2.** View of the studied caves and shelters: (**a**) southern entrance of the Reguerillo Cave; (**b**) view from inside in the Cova dos Mouros; (**c**) rock shelter of Abrigo de San Lazaro; (**d**) rock shelter of Abrigo del Molino.

Cova dos Mouros is a geological structural controlled shelter with prehistoric painting located in Baleira, Galicia (Figure 2b), northwest of Spain. These are motifs, preferably geometric, which are red in color and painted directly on the quartzite stone support. Although the site is still under study, the excavations carried out [31,32] detect traces of human presence from the Middle Neolithic (IV Millennium BC) to the Bronze Age Initial (transition III–II Millenniums BC). All this allows confirming the stylistic ascription of the paintings to what was to become known as "schematic painting".

The cave is around 20 m depth of predominantly quartzite. The geomechanical characterization was conducted both at the entrance and inside the cave.

The two shelters located in Segovia city are called Abrigo San Lazaro [33,34] (Figure 2c) and Abrigo del Molino [35] (Figure 2d). Both shelters are small cavities originated in the Cretaceous dolostones and dolomitic sandstones of the left bank of the Eresma river (Segovia, Spanish Central System piedment, Central Iberia).

The geoarchaeological research in the Abrigo del Molino and Abrigo de San Lázaro presents a rich sequence of levels that covers substantially all oxygen isotope stages 3 (OIS 3), including several levels with human occupation in both sites. These results are of great value for analyzing the end of Neanderthal human occupations in the Iberian Peninsula and to obtain contextual data which can help us to establish a chronostratigraphy in the terraces of the upper part of the Eresma River valley thanks to the palaeo-flood levels documented in the basis of the sequence.

## 3. Materials and Methods

### 3.1. Photogrammetry Modeling

#### 3.1.1. Photogrammetry SfM

Photogrammetry is a remote sensing technique that enables the extraction of a 3D geometric property from a pair or a set of images depicting a scene. The different strategies to obtain this 3D information are based on the principles of stereoscopic vision (using only two photographs) or in modern 3D reconstruction techniques using automatic correlation algorithms of images [5,15,16,24–26].

Structure from Motion (SfM) technology has recently emerged as a highly efficient alternative. Its primary advantage lies in its ability to utilize multiple overlapping photographs to determine the camera's orientation parameters, thus eliminating the need for calibration (Figure 3). To achieve this, the SfM algorithm follows these steps:

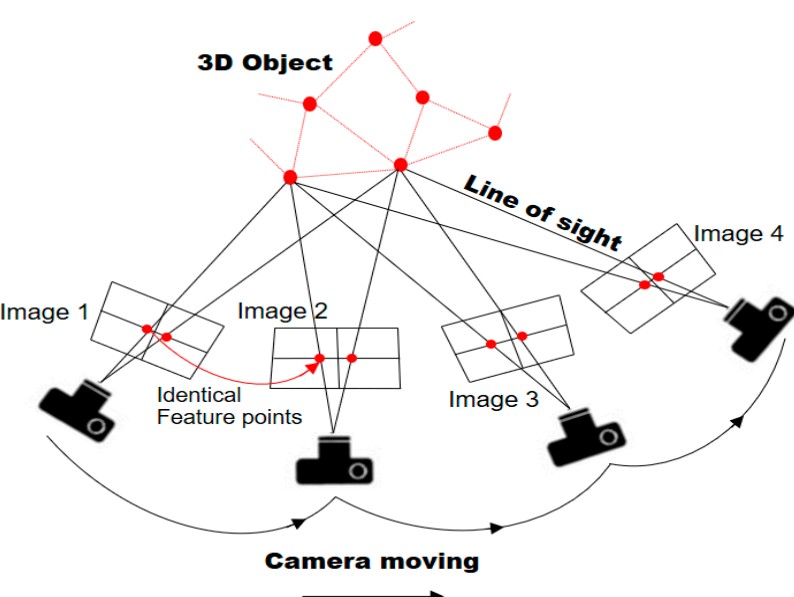

**Figure 3.** Principle of SfM photogrammetry for 3D object modeling.

(i) It detects numerous key-points within each 2D image. (ii) These key-points are then matched in overlapping images. (iii) An iterative bundle adjustment algorithm is employed to estimate the camera parameters for each image, allowing for the computation of 3D positions of these key-points and the initial creation of a scattered 3D point cloud. (iv) Subsequently, a dense 3D point cloud is generated using Multi-View Stereo (MVS) techniques, which involve the correspondence between points found in more than two images. (v) Finally, the point cloud can be scaled and oriented within a reference system by utilizing a minimum of three ground control points (GCPs) (see Section 3.1.2). These GCPs are identifiable in the photos, and their coordinates within the system are known. [25].

This technique can be applied using software packages like Agisoft Metashape or ReMake Autodesk.

In the field campaign, 89 photographs at Cova dos Mouros, 160 in the Reguerillo Cave, 159 in the Abrigo del Molino rock shelter and 84 in Abrigo de San Lazaro were taken. All

photographs were captured with the maximum possible resolution (1 MP, 3:2; effective pixels 3888 × 2592; RAW-JPEG format). These photos were gathered by an amateur digital Nikon coolpix 2800 "low-cost" camera using a constant focal distance (10.4 mm) and constant camera settings considering the normal light condition of the sites to generate a quick 3D point cloud with the Structure from Motion (SfM) methodology for each site. The variation in the number of photographs for each site is influenced by the extent and diversity of geometric irregularities present in the studied area.

To generate the 3D models of the caves and the shelters studied from the photographs taken in each site, the software Agisoft Metashape [36] was used. This software allows building 3D models by selecting the level of computation, which influences the quality of the results (Figure 4). In this work, all the 3D models were generated in high precision to obtain the quality required that allows visually observing the discontinuities sets, and the computation time for the generation of each model was about 24 h.

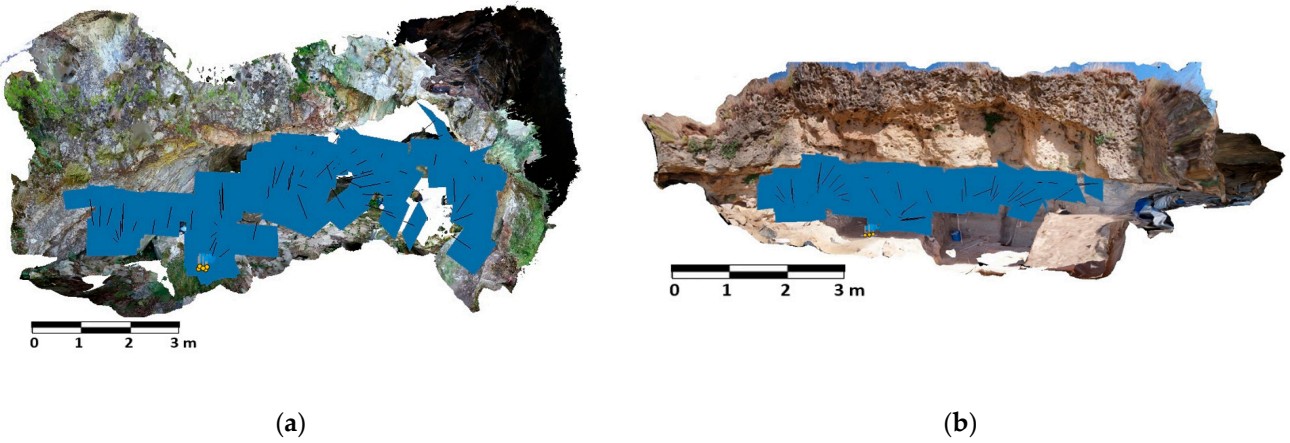

(**a**)                                                                                     (**b**)

**Figure 4.** Three-dimensional (3D) point cloud obtained with Agisoft metashape Professional. Blue rectangles represent the position of the photographs taken at (**a**) Cova dos Mouros and (**b**) Abrigo de San Lazaro.

### 3.1.2. Portable Orientation Template

The 3D cloud point generated should be correctly oriented and scaled. And to avoid the use of a topographic control device, a powerful rapid and low-cost tool "portable orientation template" has been developed [25,26]. This template is reminiscent of a traditional compass, but on a larger scale, it includes five ground control points (GCPs), 3 axes (x, y, z), among which the y-axis can be aligned to the north using a compass, and a spirit level that can be used to ensure the template's horizontal placement. With the GCP coordinates already known, the template functions as a local reference plane (Figure 5).

In the pursuit of a thorough evaluation of the quality of the point cloud 3D model, we conducted a meticulous assessment of its accuracy. This assessment involved a detailed comparison of the known, real-world coordinates of ground control points (GCPs) against the corresponding coordinates generated by the model.

Table 1 meticulously delineates the deviations in the x, y, and z axes for each GCP within every model from the test sites. These values are reasonable [24] and represent critical metrics for assessing the model's precision and accuracy.

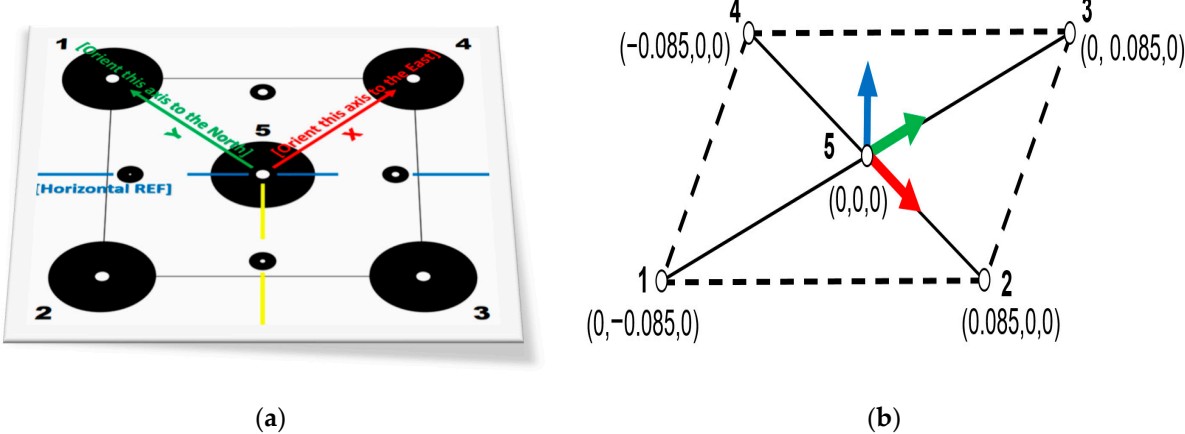

(**a**)                       (**b**)

**Figure 5.** Portable orientation template: (**a**) the template used; (**b**) GCPs coordinates in (m).

**Table 1.** Errors of (x, y, z) coordinates of GCPs within the 3D model.

|  | GCPs | x Error (mm) | y Error (mm) | z Error (mm) | Total (mm) |
|---|---|---|---|---|---|
| Reguerillo Cave | 1 | 0.985898 | −0.9926550 | 1.01412 | 1.72794 |
| | 2 | 0.1439340 | 0.664821 | −0.611245 | 0.914508 |
| | 3 | 0.146859 | −0.425902 | −0.0376319 | 0.45208 |
| | 4 | −0.0132861 | −0.138603 | −0.0762247 | 0.158737 |
| | 5 | 0.708403 | 0.892338 | −0.28902 | 1.17543 |
| Cova dos Mouros | 1 | −0.29687 | 0.08203560 | 0.6063450 | 0.680085 |
| | 2 | 0.0523924 | 0.375666 | 0.1767530 | 0.418463 |
| | 3 | 0.421436 | 0.0294192 | 0.606319 | 0.738983 |
| | 4 | −0.0453547 | −0.342616 | 0.176763 | 0.388186 |
| | 5 | −0.131602 | −0.144509 | 1.56618 | 1.57833 |
| Shlter Abrigo del Molino | 1 | 0.217393 | −0.0277667 | −0.827614 | 0.85614 |
| | 2 | 0.233013 | 0.45436 | 0.752479 | 0.909375 |
| | 3 | 0.0930127 | −0.58874 | −0.827582 | 1.01988 |
| | 4 | 0.334297 | −0.249741 | −0.441607 | 0.607571 |
| | 5 | 0.0959646 | −0.600143 | −0.733314 | 0.952434 |
| Shelter Abrigo de San Lazaro | 1 | 0.2281 | 0.919401 | −0.163856 | 0.961341 |
| | 2 | 0.446589 | 0.00941694 | 0.963992 | 1.06246 |
| | 3 | −0.589814 | −0.597718 | 0.058881 | 0.841792 |
| | 4 | 0.465558 | 0.0521444 | −0.301093 | 0.556884 |
| | 5 | 0.0741555 | −0.255769 | −0.182387 | 0.322772 |

Furthermore, Table 2 provides the calculated root mean squared (RMS) error values derived from the point cloud models. These RMS error values offer valuable insights into the overall accuracy and reliability of the models.

**Table 2.** Total RMS errors.

|  | x Error (mm) | y Error (mm) | z Error (mm) | Total (mm) |
|---|---|---|---|---|
| Reguerillo Cave | 0.550689 | 0.696308 | 0.546408 | 1.04243 |
| Cova dos Mouros | 0.239942 | 0.239578 | 0.806312 | 0.874705 |
| Shlter Abrigo del Molino | 0.215021 | 0.441898 | 0.730593 | 0.880496 |
| Shelter Abrigo de San Lazaro | 0.405363 | 0.504141 | 0.465515 | 0.796982 |

3.1.3. Analysis of the 3D Point Cloud Extraction of Discontinuities

Using the open software CloudCompare [37] and following a specific process, discontinuities sets were analyzed and determined. First, we conducted a semi-automatic analysis using the plugin Facet/fracture detection, which allows observing the plan sets

represented in different colors (Figure 6); each color represents a family of discontinuities. After that, the orientation of those discontinuities can be measured using the tool compass by point selection (Figure 6). This technique allows obtaining orientation in remote and risky zones.

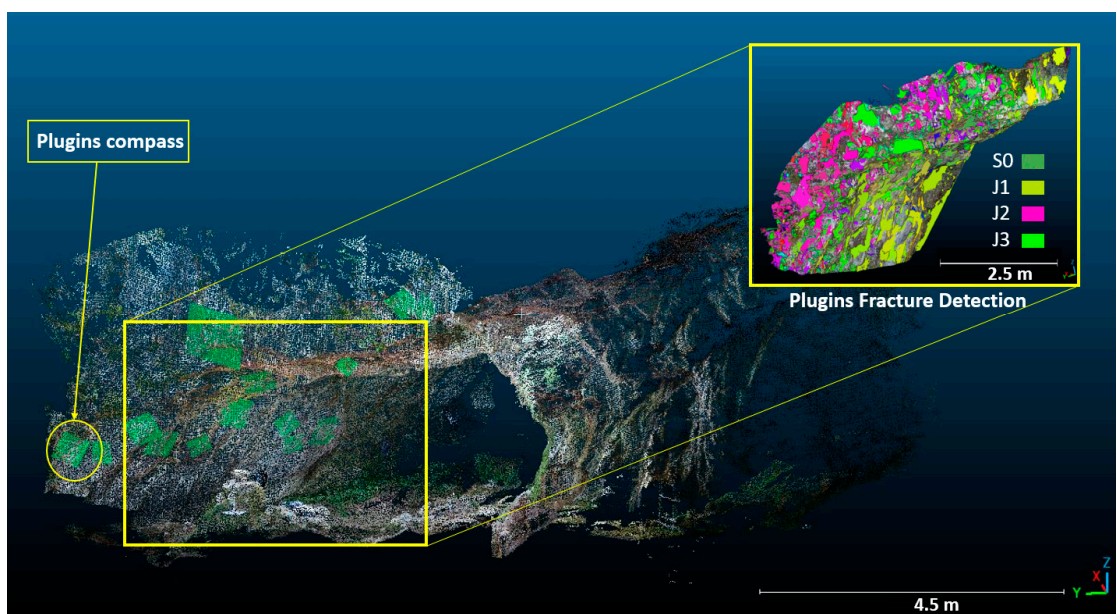

**Figure 6.** Orientation measurements and discontinuities analysis of Cova dos Mouros using Cloud-Compare software 2.12.4.

To compare and validate the data collected from the 3D point cloud with the orientation measured in the field using a compass, a pair of control planes was carefully selected at each site. These control planes, as exemplified in Figure 7, underwent manual measurement procedures facilitated by a compass. These measurements were conducted at a specific location, which was denoted by a prominently marked green reference point (Figure 7b). Simultaneously, data pertaining to the same location were acquired from the 3D point cloud. This dual approach ensures a comprehensive comparison between manual measurements and those obtained from the 3D point cloud at the identical reference point.

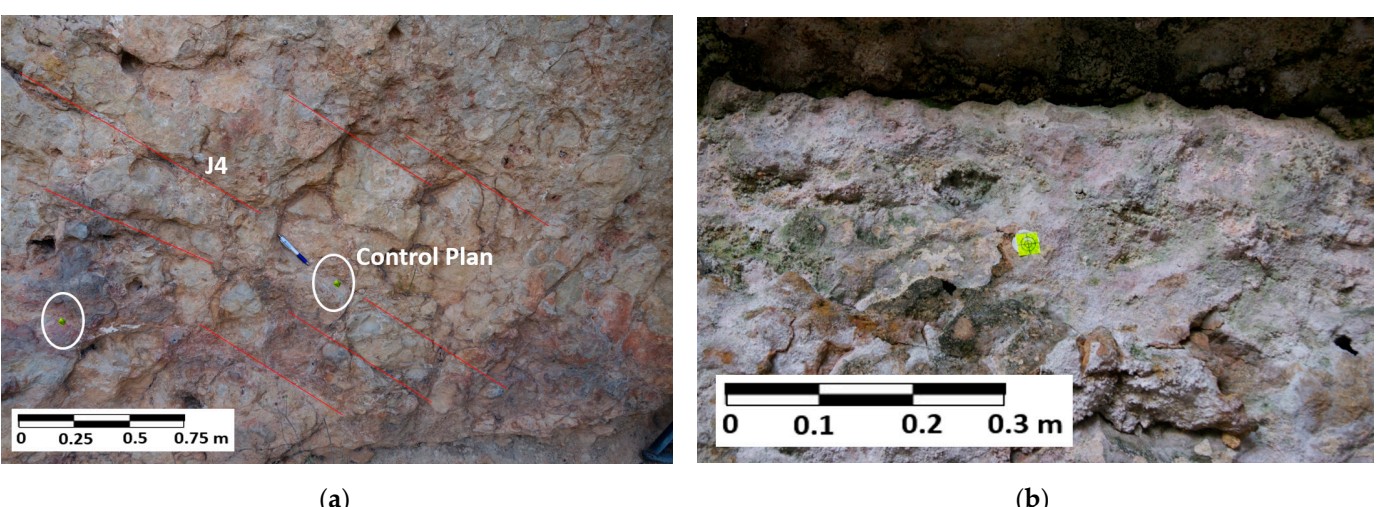

**Figure 7.** Control planes utilized for result validation (**a**) The green mark used to identify the plan to be measured; (**b**) example of 2 selected planes in Reguerillo Cave.

### 3.2. Geomechanical Station and Manual Compass Measurements

The geotechnical stability of shallow caves in jointed rock mases is controlled by the orientation and strength of discontinuities. In these cases, no tensional effects nor plastification are expected: The behavior of the terrain involves "rigid blocks" moving or shearing among themselves. This is the case of the caves and shelters of this investigation: competent jointed rock mass at very shallow depth (below 20 m overburden thickness). Generally, in rock engineering and geology, the orientation of discontinuities (joints, faults, stratification) is represented by dip angle and dip direction. The widely employed instrument to determine the orientation with respect to magnetic north is the compass [5,15], which requires access to the site (Figure 8a). However, when it comes to caves, the process raises some disadvantages: only fractures located in the lower zone can be assessed (Figure 8b). In all our study sites, we focused on obtaining measurements of plans below 2 m. We measured the discontinuity orientations using a Freiberger geological compass, and the process took approximately 30 min at each site. In the Reguerillo Cave and Cova dos Mouros, we recorded 12 and 15 measurements, respectively, while in Abrigo del Molino and Abrigo San Lazaro rock shelters, we recorded 10 and 19 measurements, respectively.

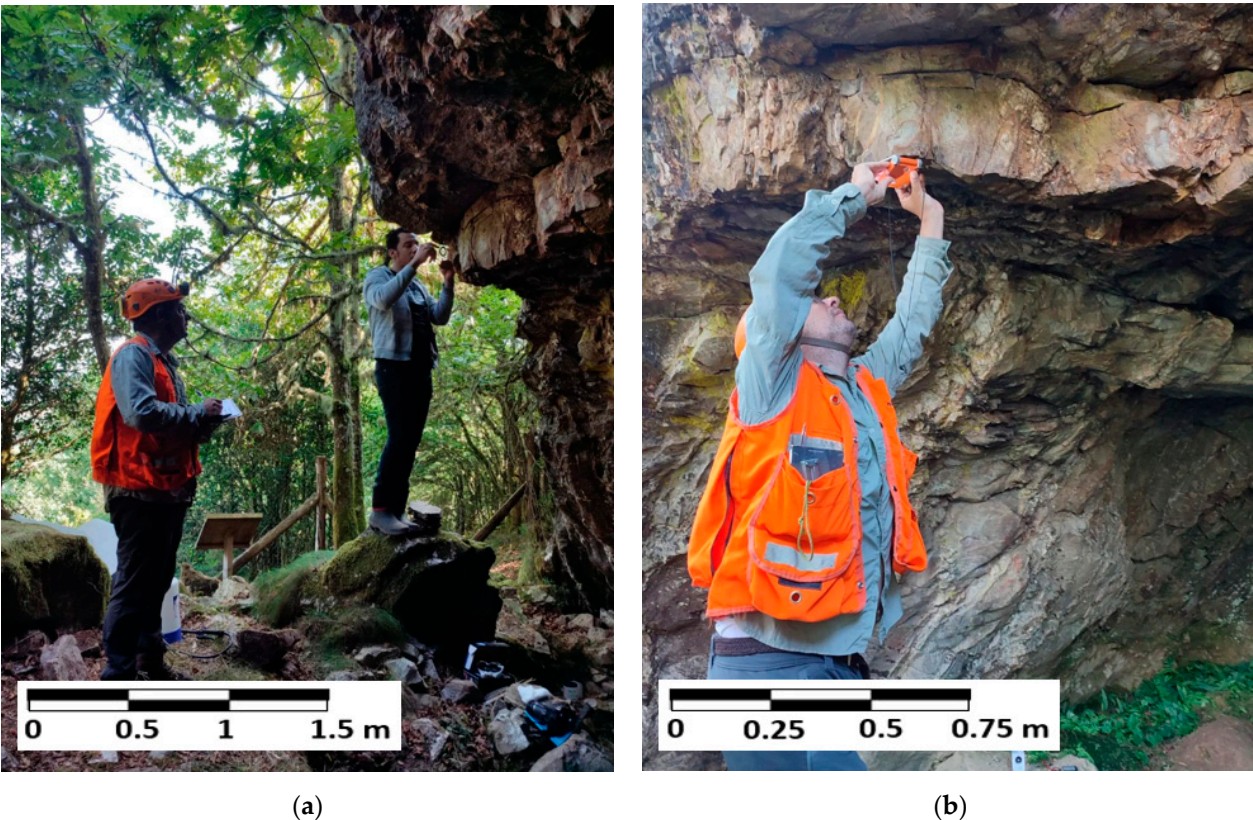

     (**a**)                                         (**b**)

**Figure 8.** Examples of difficulties collecting data in high zones by hand in Cova dos Mouros, Lugo, Galicia. (**a**) The need for at least two persons to collect data; (**b**) Difficulties in reading the compass while taking measurements of the roof and elevated areas.

To further assess the accuracy of the 3D point cloud, we initially compared the difference in measurements of the control plane (see Section 3.1.3), this comparison was initially performed in the field using a compass and subsequently within the 3D point cloud using the tool compass in Cloudcompare. Table 3 indicates a high level of similarity in the values obtained, with only a small difference observed. This variance is considered reasonable and consistent with the typical precisions acknowledged when characterizing spaces with difficult accessibility [5,15,29]. Moreover, these errors are comparable to manual measure-

ments made with a compass, which rely on the expertise and judgment of the engineers and typically range between 2 to 8 degrees on average.

**Table 3.** Errors and comparison of data measurements (manual/SfM).

| | Manual Compass | | 3D Point Cloud | | Errors | |
|---|---|---|---|---|---|---|
| | DipDir/Dip (°) | | DipDir/Dip (°) | | DipDir/Dip (°) | |
| Cova dos Mouros | J1 (074/42) | J4 (260/85) | J1 (074/46) | J4 (261/82) | J1 (000/04) | J4 (001/03) |
| Reguerillo Cave | J4 (175/40) | J4 (165/40) | J4 (179/41) | J4 (162/40) | J4 (004/00) | J4 (003/01) |
| Shlter Abrigo del Molino | E (307/45) | J3 (210/65) | E (301/48) | J3 (214/67) | E (006/03) | J3 (004/02) |
| Shelter Abrigo de San Lazaro | J1 (340/80) | S0 (065/28) | J1 (343/86) | S0 (69/34) | J1 (003/06) | S0 (04/06) |

## 4. Results

### 4.1. Reguerillo Cave

Figure 9 shows the results of the analysis conducted to identify discontinues sets within the Reguerillo Cave. Figure 9a illustrates 12 poles obtained manually by compass at the entrance of the cave, identifying four main sets (J1, J2, J3, E) with their corresponding dip/dip direction. Therefore, Figure 9b shows the results obtained from the 3D point cloud (50 poles), and it is clear that more values are shown for joints J2 and J4. We highlight the appearance of J3, which is not visible in Figure 9a. Figure 9c shows a comparison and the difference of the orientation values of the main planes: the results are highly similar with a small difference in the scattering of poles obtained. In Figure 9d, a combination of poles acquired with both techniques, manually by compass and from the 3D point cloud, is shown. This combination significantly improves the original stereogram (Figure 9a). The addition contribution of more values obtained in various locations of the cave, especially the roof, allows the observation of joint J3.

### 4.2. Cova dos Mouros Shelter

During the field campaign, two geomechanical stations (GS) were established to identify discontinuity sets within Cova dos Mouros. The results of GS2 are illustrated in Figure 10. Specifically, Figure 10a shows 15 poles obtained by manual measurements with a compass, and Figure 10b shows 50 poles obtained from the 3D point cloud. Both figures highlight the main sets (J1, J2, J3) with their corresponding dip/dip direction. Special consideration was given to joint S0, which remains unseen in Figure 10a due to the challenges in manually collecting data with a compass and difficulties in accessing its location in the cave's ceiling. The red and yellow colors indicate areas with the highest concentration of joint poles. Figure 10c displays the error and comparison of the orientation values of the main planes with the results being significantly similar with only a minor difference. Figure 10d shows a combination of poles obtained with a compass and from the 3D point cloud. This combination of data clearly improved the original stereogram (Figure 10a) with additional values.

### 4.3. Abrigo del Molino Rock Shelter

Three geomechanical stations (GS) were conducted in the Abrigo del Molino rock shelter. The results of the discontinuities collection from the main GS3 are illustrated in Figure 11 below with similar observations as before. Figure 11a depicts data obtained manually with a compass in the field, in contrast to Figure 11b, which represents data collected from the 3D point cloud. The most notable observations are J3 and J4, which are not visible in Figure 11a. Figure 11c displays the error and comparison; the results are remarkably similar with only a small difference. The combination of data significantly improved the original stereogram in Figure 11a, as evident in Figure 10d.

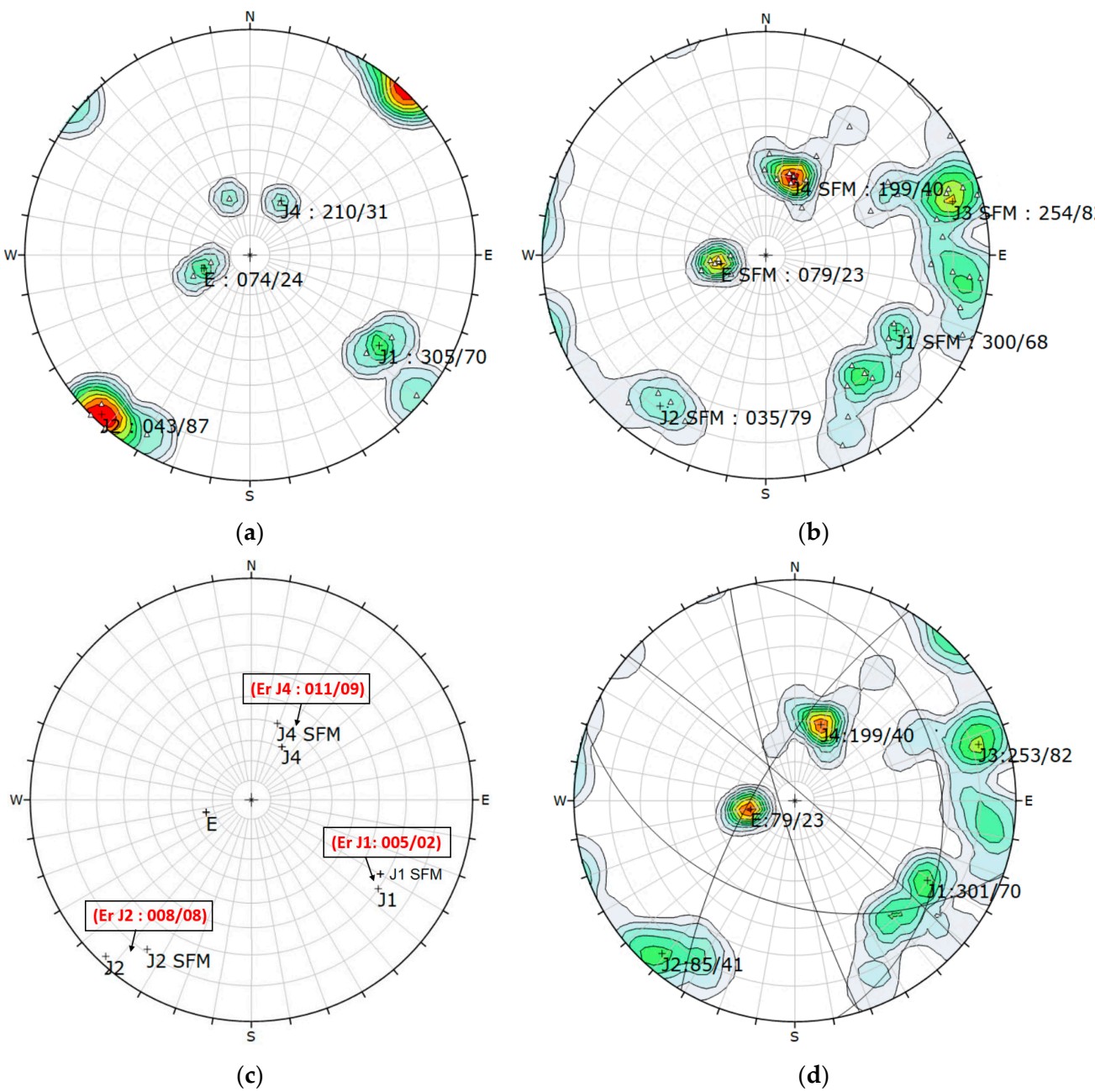

**Figure 9.** Pole concentration diagram of Reguerillo Cave using DIPS obtained from (**a**) manual measurements using a compass (n = 12 poles); (**b**) measurements acquired from 3D point cloud with CloudCompare (50 poles); (**c**) comparison of planes measured manually and from the 3D point clouds (SfM); (**d**) average orientations of the discontinuity sets derived from the combination of data presented in (**a**,**b**).

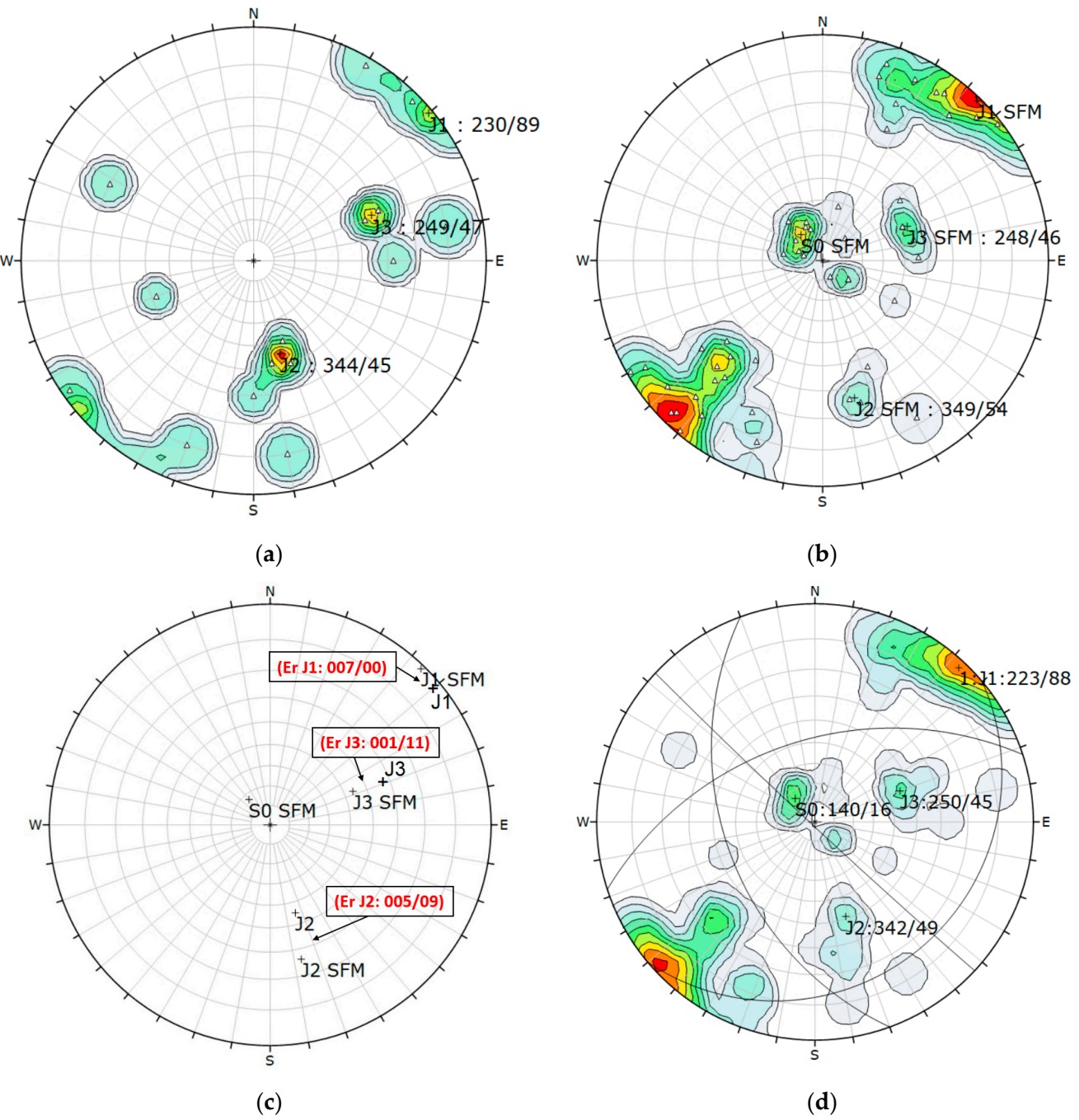

**Figure 10.** Pole concentration diagram of Cova dos Mouros GS2 using DIPS obtained from (**a**) manual measurements using a compass (n = 15 poles); (**b**) measurements acquired from 3D point cloud with CloudCompare (50 poles); (**c**) comparison of planes measured manually and from the 3D point clouds (SfM); (**d**) average orientations of the discontinuity sets derived from the combination of data presented in (**a**,**b**).

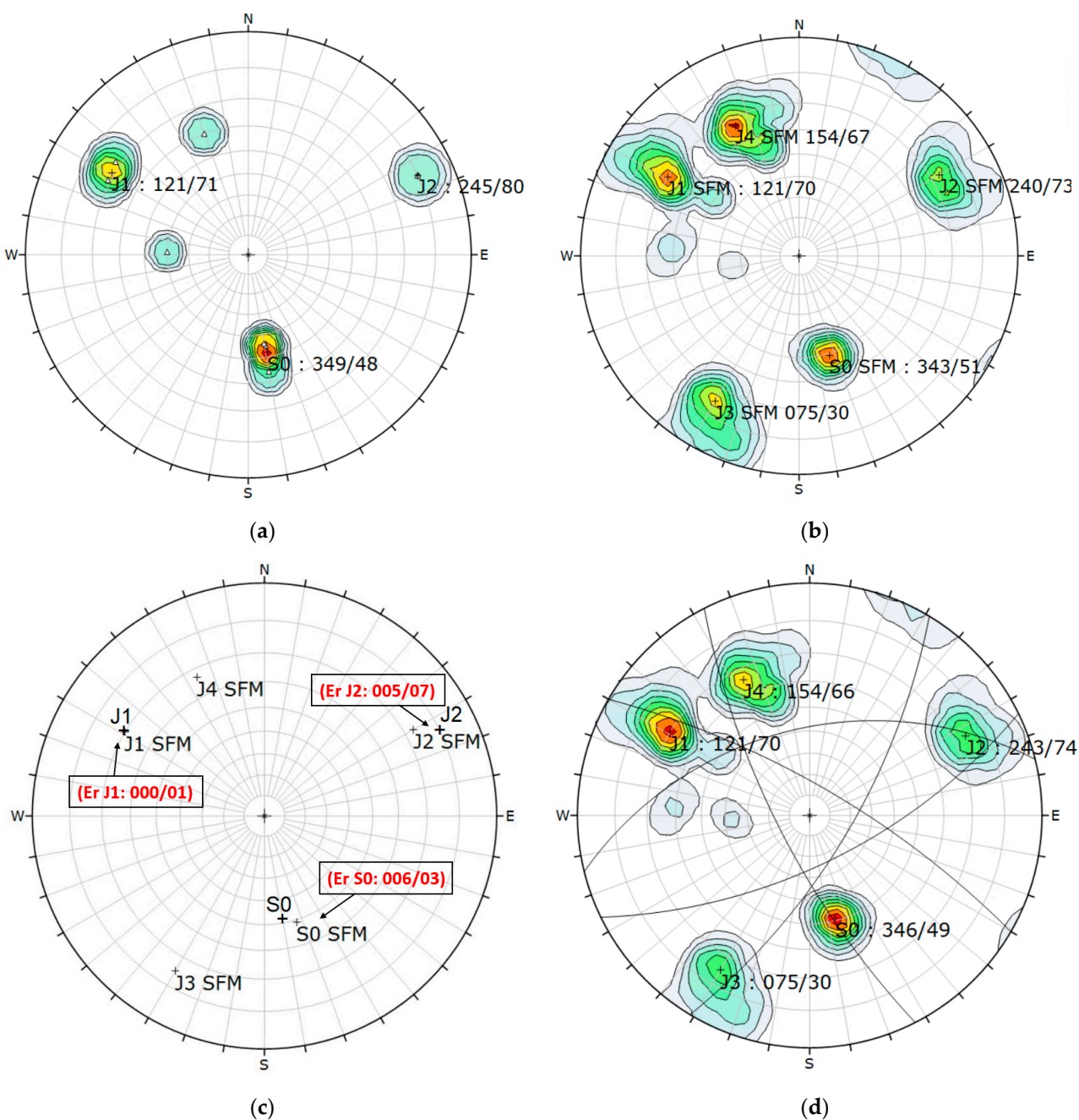

**Figure 11.** Pole concentration diagram of Abrigo del Molino rock shelter GS3 using DIPS obtained from (**a**) manual measurements using a compass (n = 10 poles); (**b**) measurements acquired from 3D point cloud with CloudCompare (50 poles); (**c**) comparison of planes measured manually and from the 3D point clouds (SfM); (**d**) average orientations of the discontinuity sets derived from the combination of data presented in (**a**,**b**).

### 4.4. Abrigo de San Lazaro Rock Shelter

The results of the discontinuities collection in Abrigo de San Lazaro are illustrated in Figure 12. We note similar observations as in the previous sites. Figure 12a showcases data obtained manually with a compass in the field, revealing only two visible joints (J1 and J2) and some dispersed poles due to challenges in manually collecting data. The most significant observation is the appearance of S0 in Figure 12b (data collected from the 3D point cloud). The errors illustrated in Figure 12c are minimal and reasonable. Lastly, the combination of data results in a rich stereogram (Figure 12d).

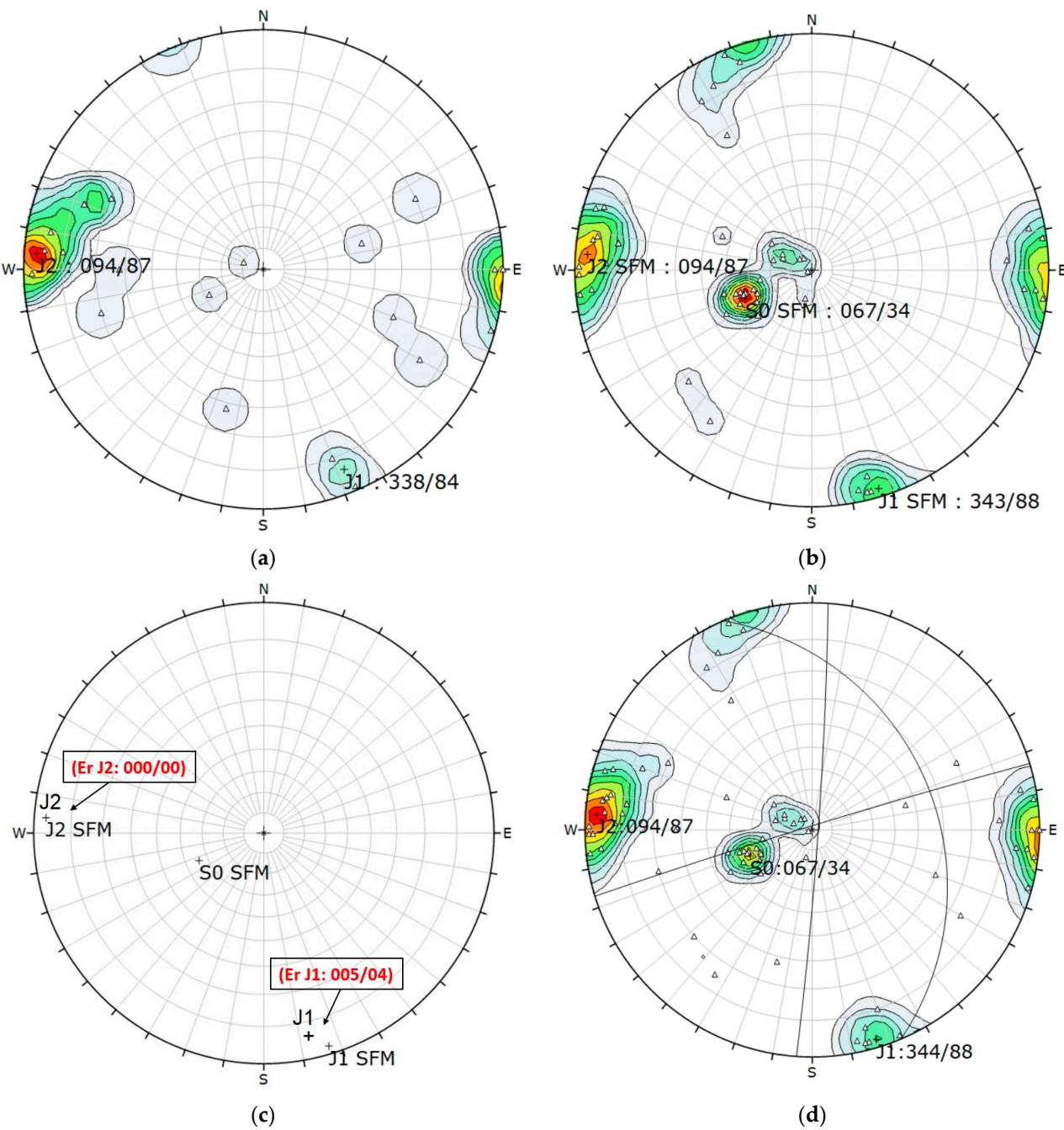

**Figure 12.** Pole concentration diagram of San Lazaro shelter obtained from (**a**) manual measurements using a compass (n = 19 poles); (**b**) measurements acquired from 3D point cloud with CloudCompare (50 poles); (**c**) comparison of planes measured manually and from the 3D point clouds (SfM); (**d**) average orientations of the discontinuity sets derived from the combination of data presented in (**a**,**b**).

Table 4 summarizes the orientation values obtained for each site. The data were collected manually via a compass in the field and from the 3D point cloud. Additionally, the table presents the differences and errors in data between both methods.

**Table 4.** Orientation values obtained of each site.

| Set | Compass DipDir/Dip (°) | Data Collection 3D Point Cloud DipDir/Dip (°) | Errors DipDir/Dip (°) | Combination DipDir/Dip (°) |
|---|---|---|---|---|
| | | Reguerillo Cave | | |
| J1 | 305/70 | 300/68 | 005/02 | 301/70 |
| J2 | 043/87 | 035/79 | 008/08 | 041/85 |
| J3 | Difficult to be measured | 254/82 | / | 254/83 |
| J4 | 210/31 | 199/40 | 011/09 | 199/40 |
| E | 074/24 | 079/23 | 005/01 | 079/23 |
| | | Cova dos Mouros GS2 | | |
| S0 | Difficult to be measured | 140/17 | / | 140/16 |
| J1 | 230/89 | 223/89 | 007/00 | 223/88 |
| J2 | 344/45 | 349/54 | 005/09 | 342/49 |
| J3 | 249/57 | 248/46 | 001/11 | 250/45 |
| | | Abrigo del Molino | | |
| S0 | 349/48 | 343/51 | 006/03 | 346/49 |
| J1 | 121/71 | 121/70 | 000/01 | 121/70 |
| J2 | 245/80 | 240/73 | 005/07 | 243/74 |
| J3 | Difficult to be measured | 075/30 | / | 075/30 |
| J4 | Difficult to be measured | 154/67 | / | 154/66 |
| | | Abrigo de San lazaro GS3 | | |
| J1 | 338/84 | 343/88 | 005/04 | 344/87 |
| J2 | 094/87 | 094/87 | 000/00 | 094/87 |
| S0 | Difficult to be measured | 067/34 | / | 067/34 |

## 5. Discussion

The orientation of the discontinuities has been determined using two methods: manual measurements via a compass and from the 3D point cloud using CloudCompare software. Two planes have been selected (see example in Figure 7) for data validation, and their measurements are illustrated in Table 1. The dip and dip direction of each plan are highly consistent with an error of less than 6 degrees at the four test sites.

The data obtained from manual compass measurements, as shown in Figures 9a, 10a, 11a and 12a, resulted in low data stereograms due to the lack of measurements in some inaccessible (above 2 m) and risky zones at all sites. In the stereogram of the Reguerillo Cave, four main sets are represented, while there are three in Cova dos Mouros, three in Abrigo del Molino, and two in Abrigo de San Lazaro. Conversely, the data obtained from the 3D point cloud are significantly better at all sites, resulting in rich stereograms. Particular attention was given to the appearance of some joints, as seen in Figures 9b, 10b, 11b and 12b: joint J3 in the stereogram of the Reguerillo Cave, joint S0 in Cova dos Mouros, joints J3 and J4 in Abrigo del Molino, and joint S0 in Abrigo de San Lazaro. These new joints are located in high inaccessible or risky zones; for instance, joint J3 in the Reguerillo Cave is on the front wall at approximately 3m high, and joint S0 in Cova dos Mouros is in the roof of the cave.

A comparison of the data illustrated in Figures 9c, 10c, 11c and 12c shows a small difference in the location of pole concentration with coherent results within a 10-degree margin in the worst case [15].

By combining the data presented in Figures 9d, 10d, 11d and 12d, it becomes evident that measurements obtained from different methods complemented each other, resulting in a more realistic stereogram representation.

The consistency of the data in Table 2 is deemed satisfactory and falls within the margin of measurement variability (uncertainty) associated with manual orientation measurement with a compass [15,26,38,39].

The proposed methodology offers several advantages over alternative approaches. Firstly, it requires readily accessible and relatively inexpensive equipment, such as a reasonably good camera. Furthermore, the personnel responsible for taking the pictures do not need advanced photography skills, unlike the use of UAVs (Unmanned Aerial Vehicles) or other methods like Lidar and TLS.

Numerous authors have underscored the positive impact of Structure from Motion (SfM) techniques, regardless of the equipment used, ranging from digital low-cost cameras [15,24,26] to UAVs (Unmanned Aerial Vehicles) [38,39]. It is a well-established fact that SfM provides a more realistic characterization of discontinuity orientation compared to traditional methods like compass clinometers. As such, SfM is expected to take on a central role in shaping the future of surveying techniques, becoming an indispensable asset for researchers, professionals, and diverse industries.

The open-source software CloudCompare offers significant benefits for the visualization and analysis of 3D remote sensing rock models. This program allows users to view point clouds from various angles, enabling comprehensive observation and the interpretation of rock discontinuities.

## 6. Conclusions

The objective of this study was to compare two methods of obtaining joint sets: manual field data collection using a compass and remote sensing data acquisition through photogrammetry with the assistance of a point cloud program. We collected significant structural data from various caves and shelters in a quick and straightforward manner, aiming to validate the methodology for remote locations.

The information was gathered using different techniques (manual data collection with a compass and Structure from Motion (SfM)), neither of which requires topographic precision. Therefore, comparing the results with values obtained from a high-precision technique was not deemed necessary, as it was not part of the study's focus.

To achieve this, control points were employed in each site. These planes are measured manually with a compass at one point identified by a green mark and from the 3D point cloud in the same point. We used a portable orientation template, equipped with five strategically placed ground control points (GCPs), and it encompasses three axes (x, y, z). The y-axis, crucial for precise alignment with the north, can be conveniently oriented using a conventional compass. To guarantee the template's level placement, a spirit level serves as a dependable tool, which proved to be a practical tool that expedited the process and replaced the need for topographic control devices.

The study primarily focused on comparing the differences and similarities in the data obtained from both techniques, including dip and dip direction measurements.

Manual data collection with a compass does have limitations, as the process relies on visual examination and manual measurement, and there are access limitations to roofs and walls for obtaining structural data. In our cases, compass data collection was restricted to lower and safe areas, introducing bias in the resulting diagrams.

This study highlighted the value of combining manually collected data with a compass and data extracted from 3D point clouds using SfM. The manual data collection and visual analysis allowed for the prior recognition of planes controlling stability. It also allows gathering data on the properties of joints that we cannot obtain remotely. By combining compass and remote data, the analysis became more objective and of better quality with the added benefit of obtaining other pole clouds for planes that were not accessible manually (Figures 9, 11 and 12).

The validity of the 3D point cloud was established through a comparison of orientations measured by both techniques (Figure 7 and Table 3). The results demonstrated coherence between compass measurements and orientations derived from the 3D point cloud. In the most unfavorable case, the orientation variation was under 10° for the pole vector but on average around 5°. Structure from Motion photogrammetry proved to

be a highly effective method for obtaining structural data in inaccessible areas of caves and shelters.

**Author Contributions:** Conceptualization, L.J.B. and A.B.; methodology, A.B.; software, A.B.; validation, S.S.D., L.J.B., M.d.A.-H., F.C.-R. and D.Á.-A.; formal analysis, A.B., S.S.D., L.J.B., M.d.A.-H., F.C.-R. and D.Á.-A.; investigation, A.B., S.S.D., L.J.B., M.d.A.-H., F.C.-R. and D.Á.-A.; resources, A.B., S.S.D., L.J.B., M.d.A.-H., F.C.-R. and D.Á.-A.; data curation, A.B., L.J.B. and S.S.D.; writing—original draft preparation, A.B., S.S.D., L.J.B., M.d.A.-H., F.C.-R. and D.Á.-A.; writing—review and editing, A.B., S.S.D., L.J.B., M.d.A.-H., F.C.-R. and D.Á.-A. All authors have read and agreed to the published version of the manuscript.

**Funding:** This research received no external funding.

**Data Availability Statement:** Data are contained within the article.

**Conflicts of Interest:** The authors declare no conflicts of interest.

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
