# Peer review of "Structural and Geomechanical Analysis of Natural Caves and Rock Shelters: Comparison between Manual and Remote Sensing Discontinuity Data Gathering"

_remotesensing, doi:10.3390/rs16010072_

Round 1

Reviewer 1 Report (Previous Reviewer 1)

Comments and Suggestions for Authors

Dear authors,

I think that you did an excellent job with the latest corrections. Just a few remarks to include in your manuscript:

- There are two Figure 1. The first one has no geological legend. The second below is deformed. 

- Additionally, I recommend to refer in line 54 the following reference:

Fernández-Lozano, J., Gutiérrez-Alonso, G., Ruiz-Tejada, M. Á., & Criado-Valdés, M. (2017). 3D digital documentation and image enhancement integration into schematic rock art analysis and preservation: The Castrocontrigo Neolithic rock art (NW Spain). Journal of Cultural Heritage26, 160-166.

Best regards

Author Response

Review Report (Reviewer 1)

Dear authors,

I think that you did an excellent job with the latest corrections. Just a few remarks to include in your manuscript:

- There are two Figure 1. The first one has no geological legend. The second below is deformed.

- Additionally, I recommend to refer in line 54 the following reference.

Reply:

Dear Reviewer,

Thank you for your positive feedback and acknowledgment of the recent corrections made to the manuscript. We appreciate your review. Based on your remarks, I will address the following points in the manuscript:

  1. As the figure is large and comprises three different regions, it is difficult to add a separate legend for geological context. For this reason, we included it inside Figure 1b. Each color has a defined meaning; we believe this was the best possible approach, and the second figure has been deleted.
  2. We would like to inform you that we have successfully incorporated the suggested citation into line 54 of the manuscript.

Thank you for your continued support and constructive feedback.

Best regards,

Reviewer 2 Report (Previous Reviewer 4)

Comments and Suggestions for Authors

Dear authors,

It is the second time I read your contribution, and I consider this new version improved. However, although it is improved, its present form is too far to be published. Your chapter 4 needs careful rewriting (see annotated version). This rewriting is not a trivial work, and given it, I conclude on rejection, at present, and resubmission. Overall, I consider the effort as worth publishing after major revision. 

Comments on the Quality of English Language

Quality of English language is low particulalry the new version chapter 4. Thorough rewriting.

Author Response

Review Report (Reviewer 2)

Dear authors,

It is the second time I read your contribution, and I consider this new version improved. However, although it is improved, its present form is too far to be published. Your chapter 4 needs careful rewriting (see annotated version). This rewriting is not a trivial work, and given it, I conclude on rejection, at present, and resubmission. Overall, I consider the effort as worth publishing after major revision.

Quality of English language is low particularly the new version chapter 4. Thorough rewriting.

Reply:

Dear Reviewer,

We appreciate your thorough review and the time you've dedicated to evaluating the manuscript. We are grateful for your acknowledgment of the improvements in the second version. We understand your concerns regarding the current state of Chapter 4 and the need for careful rewriting, as indicated in the annotated version. What happened to our manuscript is that we encountered a problem when saving the file as a PDF. The corrections didn't appear, resulting in the duplication of many words and sentences, and we didn't notice that. In the new version, we corrected all the errors.

We have retained the previous version of the PDF, which includes responses to all comments. The new word version is cleaner and does not provide detailed explanations for the modifications.

Thank you for your feedback regarding the quality of the English language in the new version of Chapter 4. We appreciate your concern and suggestion. In response, we have rewritten the entire chapter, ensuring that the issues We mentioned will not be repeated in the new version.

Round 2

Reviewer 2 Report (Previous Reviewer 4)

Comments and Suggestions for Authors

Given the changes from the first to this second version the contribution is improved. So, I consider it is acceptable for publication.

Comments on the Quality of English Language

The written English is well understood. However a final editing before the publication is appropriate.

This manuscript is a resubmission of an earlier submission. The following is a list of the peer review reports and author responses from that submission.

Round 1

Reviewer 1 Report

Comments and Suggestions for Authors

Dear Authors,

Although his article shows a technical approach of some widely used applications in the 3D documentation of complex natural cave environments, the application of automatic measurement techniques does not represent any progress concerning what has been experienced in discontinuity analysis in recent years. They have, for example, applications to use, such as those published by authors such as Riquelme et al. or Wu et al. The latter is in this same magazine. Applying a tool like the one they use in Cloud Compare does not represent a significant advance concerning existing new techniques. It is widely used in many research fields, including yours. On the other hand, the results of the measurements they carry out will depend, to a large extent, on the accurate reconstruction of the natural environments, so without a detailed list of the errors and deformations produced during modelling with the Structure from Motion technique, they could get orientations different from the real ones, since the planes are not in their proper position. Despite all this, and even though they have carried out a comparison between two alternative methods, the potential interest of this type of study is reduced by the number of processing options and the little information that can be obtained from it compared to other analysis tools (see comment above) which is limited for a magazine of these characteristics.

Additionally, the literature regarding this methodology is scarce, and recent works are cited neither in ​​the archaeological part of their work nor in the geotechnical part (see some important examples below). Therefore, the discussion of the results remains weak, which only compares what has already been shown in the results. To present it in other publications, I recommend you strengthen the discussion since there is no significant difference between the results and discussion sections. Moreover, the initial figure 1 must be improved, and the geological units must be correctly placed. It is difficult to differentiate between them, change textures, or add others. See also minor mistakes. Likewise, the bibliography has not been drawn up homogeneously, and there are appreciable differences among them. Other minor things are attached in the pdf.

I hope these comments will help you improve the manuscript.

Best regards

Suggested references:

Lee, Y. K., Kim, J., Choi, C. S., & Song, J. J. (2022). Semi-automatic calculation of joint trace length from digital images based on deep learning and data structuring techniques. International Journal of Rock Mechanics and Mining Sciences149, 104981.

Yi, X., Feng, W., Wang, D., Yang, R., Hu, Y., & Zhou, Y. (2023). An efficient method for extracting and clustering rock mass discontinuities from 3D point clouds. Acta Geotechnica, 1-19.

Menegoni, N., Giordan, D., Perotti, C., & Tannant, D. D. (2019). Detection and geometric characterization of rock mass discontinuities using a 3D high-resolution digital outcrop model generated from RPAS imagery–Ormea rock slope, Italy. Engineering geology, 252, 145-163.

Ferrero, A. M., Forlani, G., Roncella, R., & Voyat, H. I. (2009). Advanced geostructural survey methods applied to rock mass characterization. Rock Mechanics and Rock Engineering, 42(4), 631-665.

Reviewer 2 Report

Comments and Suggestions for Authors

This manuscript tries to comparison between manual and remote sensing discontinuities data gathering. I just have some points to raise a suggestion.

1.      There are no innovative scientific points in the content of this manuscript, and most of them are based on the development of relatively mature remote sensing technology. The author also did not detail the specifications and parameters of the instruments and equipment used. The content of the technical report is a bit weak.

2.      The geological map and index map in Figure 1 are too rough and simple, it is recommended to use the usual representation of geological maps.

3.      The resolution of several figures is not good or the content text is not easy to recognize. It is recommended to think about the focus of the performance.

4.      Some figures also lack important components or are too small to be read easily, such as scale, north, direction.

Comments on the Quality of English Language

1.Legend texts in geological maps should use geological professional terms.

2.Proper nouns such as place names should be the same in the text and figure.

Reviewer 3 Report

Comments and Suggestions for Authors

Dear Editor,

I am writing to provide my review of the manuscript titled "Structural and geomecanical analysis of natural caves and rock shelters: Comparison between manual and remote sensing discontinuities data gathering", which I had the opportunity to evaluate as a reviewer.

Firstly, I would like to express my appreciation for the authors' research efforts. The topic addressed in the manuscript is indeed intriguing and has the potential to contribute significantly to the scientific community. The results and findings presented are promising and hold valuable insights.

However, after a thorough review, I recommend that the manuscript undergoes careful revision before it can be considered suitable for publication. The article requires attention, I find the manuscript promising, but a thorough revision is necessary to address the comment points inserted in the pdf file. 

With the suggested improvements, the manuscript has the potential to make a valuable contribution to the field and align more closely with the standards of the Remote Sensing journal.

I look forward to reviewing the revised version of the manuscript and witnessing the progress made by the authors. 

Thank you for considering my review.

Sincerely,

Anonymous Reviewer

Reviewer 4 Report

Comments and Suggestions for Authors

Dear authors,It is a study that tries to solve a problem, which is the analysis of joints in caves. By definition, it is a difficult problem, and I think the manuscript that tries to address it is worth publishing. However, significant parts of the methodology are obscure in the current version. Several other problems are highlighted in the annotated version of your manuscript. So, the value of the work is difficult to estimate. For this reason, I propose a major revision and resubmission.   

Comments on the Quality of English Language

Editting of the text is for sure needed!
